# Review of Experimental Studies to Improve Radiotherapy Response in Bladder Cancer: Comments and Perspectives

**DOI:** 10.3390/cancers13010087

**Published:** 2020-12-30

**Authors:** Linda Silina, Fatlinda Maksut, Isabelle Bernard-Pierrot, François Radvanyi, Gilles Créhange, Frédérique Mégnin-Chanet, Pierre Verrelle

**Affiliations:** 1French League Against Cancer Team, CNRS UMR144, Curie Institute and PSL Research University, 75005 Paris, France; isabelle.bernard-pierrot@curie.fr (I.B.-P.); francois.radvanyi@curie.fr (F.R.); 2CNRS UMR 9187, INSERM U1196, Curie Institute, PSL Research University and Paris-Saclay University, Rue H. Becquerel, 91405 Orsay, France; fatlinda.maksut@curie.fr (F.M.); frederique.megnin@curie.fr (F.M.-C.); 3Radiation Oncology Department, Curie Institute, 75005 Paris, France; gilles.crehange@curie.fr; 4Clermont Auvergne University, 63000 Clermont-Ferrand, France

**Keywords:** bladder cancer, radiotherapy, radiosensitisation, molecular subtypes, preclinical studies, bladder cancer cell lines

## Abstract

**Simple Summary:**

Bladder cancer is a major global health problem. Bladder removal surgery is the standard treatment for muscle-invasive bladder cancer (25% of all bladder cancer), but this treatment negatively affects the quality of life, especially for elderly and frail patients. Tumour resection followed by combination of radiotherapy and chemotherapy has emerged as a promising bladder preserving strategy. However, this strategy is unable to avoid radiation-related bladder side effects. Therefore, it is of great interest to discover novel strategies radiosensitising tumours while sparing normal bladder tissue. In this review, we analysed the experimental studies of radiosensitising strategies in bladder cancer and provided suggestions to improve forthcoming studies.

**Abstract:**

Bladder cancer is among the top ten most common cancer types in the world. Around 25% of all cases are muscle-invasive bladder cancer, for which the gold standard treatment in the absence of metastasis is the cystectomy. In recent years, trimodality treatment associating maximal transurethral resection and radiotherapy combined with concurrent chemotherapy is increasingly used as an organ-preserving alternative. However, the use of this treatment is still limited by the lack of biomarkers predicting tumour response and by a lack of targeted radiosensitising drugs that can improve the therapeutic index, especially by limiting side effects such as bladder fibrosis. In order to improve the bladder-preserving treatment, experimental studies addressing these main issues ought to be considered (both in vitro and in vivo studies). Following the Preferred Reporting Items for Systematic Reviews and Meta-Analyses (PRISMA) guidelines for systematic reviews, we conducted a literature search in PubMed on experimental studies investigating how to improve bladder cancer radiotherapy with different radiosensitising agents using a comprehensive search string. We made comments on experimental model selection, experimental design and results, formulating the gaps of knowledge still existing: such as the lack of reliable predictive biomarkers of tumour response to chemoradiation according to the molecular tumour subtype and lack of efficient radiosensitising agents specifically targeting bladder tumour cells. We provided guidance to improve forthcoming studies, such as taking into account molecular characteristics of the preclinical models and highlighted the value of using patient-derived xenografts as well as syngeneic models. Finally, this review could be a useful tool to set up new radiation-based combined treatments with an improved therapeutic index that is needed for bladder preservation.

## 1. Introduction

While radical cystectomy has taken the central place in the treatment of muscle-invasive bladder cancer (MIBC) in recent decades, radiation-based treatments have also been investigated. Radiotherapy (RT) alone with curative intent for MIBC was extensively used in the 1950s through the 1980s. From 1981 to 1985, the addition of concurrent chemotherapy to RT was investigated. The National Bladder Cancer Group first used cisplatin as a radiosensitiser for MIBC patients who were ineligible for cystectomy and observed high complete response and survival rates, which consequently encouraged further studies [1] (see Appendix A).

Housset and colleagues first reported promising findings using 5-fluorouracil (5-FU) + cisplatin combination as a radiosensitiser in MIBC [2]. Following further studies, it became evident that the concurrent chemoradiotherapy (CCRT) improves locoregional disease control in MIBC as compared to RT alone [3,4,5]. However, despite the existing volume of research, there remains no standard procedure of CCRT regimen. Although different chemotherapy (CT) agents have been investigated, most evidence exists for cisplatin [3] or mitomycin C + 5-FU, [4] and more recently for gemcitabine [6]. In addition, other approaches have been explored such as the use of nicotinamide and carbogen to fight hypoxia-related radioresistance [7]. Mitomycin C + 5-FU is a very effective radiosensitising combination that has improved clinical outcomes in head and neck and anal cancers [8,9,10]. Although cisplatin + 5-FU is widely delivered, mitomycin C + 5-FU is also a common combination particularly for frailer and elderly bladder cancer (BCa) patients, given the absence of nephrotoxicity when compared to platinum drugs [4,11].

With the advances of the cystectomy techniques, radical cystectomy with pelvic lymphadenectomy and cisplatin-based CT has become the gold standard treatment for patients with MIBC. RT can be considered as an adjuvant therapy following radical cystectomy in patients with pathological high-risk of loco regional relapse (i.e., pT3-4, positive nodes, positive surgical margins), but the pelvic toxicity remains significant despite the advances in RT such as intensity-modulated radiation therapy. This management approach is supported by numerous renowned organisations, such as the National Comprehensive Cancer Network in the United States [12], as well as by the European Association of Urology [13]. In fact, the latter has made strong recommendation to use cisplatin based neoadjuvant CT before radical cystectomy for treating MIBC (T2-T4aN0M0) and high-risk non-muscle invasive bladder cancer. Level 1a evidence supports that neoadjuvant cisplatin-based CT increases survival at 6 years by 8% [13,14]. Although post-radical cystectomy history can be associated with increased risk of infection, extensive bleeding, affected sexual function and quality of life, it achieves locoregional control and results in 60% of the overall 5-year survival [15,16]. The absence of prospective randomised studies has impeded comparison of radical cystectomy versus other forms of therapy [17]. The treatment choice for MIBC between radical cystectomy versus bladder preservation largely depends on the specialist expertise in the treatment centre and often varies among countries.

Although RT has been used in bladder cancer (BCa) treatment since the 1950s, there is a relatively low number of experimental studies on the topic of radiosensitisers in BCa compared to other cancer types. The first study identified was in the 1979 [18]. Altogether, we identified 85 studies investigating RT in BCa experimental models published between 1979 and October 2020.

## 2. Radiotherapy as Part of Bladder Preserving Treatment in Clinics

In the last decade, trimodality treatment consisting of maximal transurethral resection of bladder tumour (TURBT) coupled with CT has emerged as a bladder sparing treatment either driven by patients’ choice or due to the patients’ ineligibility for radical cystectomy. In most of the CCRT protocols, including the pioneering study of Housset et al. [2], following cystoscopic evaluation of the initial CCRT response, good responders complete the CCRT schedule. Bladder preservation outcomes heavily depend on tumour response to CCRT (reviewed by [19]) and in the case of poor response, radical cystectomy is planned [2]. A standard RT schedule consists of external beam radiation therapy (EBRT) to the bladder and limited pelvic lymph nodes with an initial dose of 40–46 Gy, with a boost to the whole bladder of 14–20 Gy, with a total dose of 60–66 Gy [20] with conventional fractionation. Partial bladder irradiation remains controversial [21] as well as a tumour dose escalation which is still investigational [22]. Moderate hypofractionation is a well-tolerated option for frailer and elderly patients, even in combination with CT [23]. In addition, RT has been recently successfully combined with several immunotherapy agents in clinical trials for metastatic BCa [24].

## 3. Limitations of Use of RT in MIBC in Clinics

There are three main limitations of using CCRT in MIBC. Firstly, there is a significant risk of pelvic recurrence (25–50%) [17]. Secondly, CCRT treatment may create damage to the bladder wall resulting in undesirable toxicity. Late toxicity is characterised by replication of the injured vascular endothelial cells and connective tissue, but failure of regeneration, and may result in fibrosis, which can lead to the need of ultimate cystectomy [25]. Presently, the underlying reasons of including whole bladder in the clinical target volume (CTV) are that the irradiation field is difficult to be adjusted to concentrate on the bulk tumour and due to the high risk of spread of bladder tumours within the urothelium layers. It is proving problematic to reliably and accurately define the CTV exact position for the RT delivery as the bladder volume is continuously changing with the level of urine and post-void residual volume [26]. Nevertheless, it is important to emphasise that modern radiotherapy techniques such as image-guided radiation therapy, intensity-modulated radiation therapy and volumetric-modulated arc therapy have significantly advanced and improved the sparing of pelvic organs, especially small intestines, while better targeting delivery to the bladder (reviewed by [23]).

Thirdly, at the present time there is a lack of validated biomarkers predicting tumour response to CCRT [27,28,29]. Several candidates from the DNA damage response (DDR) pathways have been investigated [30,31,32]. Unfortunately, even the most promising biomarkers, such as double strand break repair nuclease MRE11 (MRE11), have failed to generate reproducible data. In the latest multicentre collaborative effort to validate MRE11 as a biomarker, the immunohistochemistry scoring results varied considerably and failed to attain a reliable dataset [33]. Finally, a complete or near complete response assessed by cystoscopy after 4–5 weeks induction phase remains as the only reliable predictor of treatment outcome [34,35]

## 4. MIBC Molecular Subtypes as Biomarkers for CCRT Response

### 4.1. BCa Tumour Subtypes

MIBC can be classified into molecular subtypes by transcriptome profiling, thus allowing patient stratification to consider different therapeutic options. However, MIBC subtyping is still not included in routine clinical practice due to several classifications existing simultaneously in the last decade.In 2020, a consensus on MIBC subtypes has been reached [36], giving hope for a rapid translation into clinics. Furthermore, a single-sample classifier has been established enabling to assign a consensus class label to a tumour sample’s transcriptome. There are six biologically relevant consensus molecular classes, namely, luminal papillary, luminal non-specified, luminal unstable, stroma-rich, basal/squamous and neuroendocrine-like [36]. Among the luminal subtypes, the most represented is the luminal papillary subtype (24% of MIBC), the other two luminal subtypes representing 15% (for the luminal unstable subtype) and 8% (for the luminal non-specified subtype) of MIBC (Figure 1a). The other most frequent subtype is the basal/squamous subtype representing 33% of MIBC. Further, 15% of MIBC represent stroma-rich and 3% neuroendocrine-like subtype (Figure 1a) [36].

Several retrospective studies have highlighted the clinical significance of molecular stratification of MIBC suggesting that responses to treatment could be predicted by tumour subtyping [37,38,39,40]. However, there was a lack of association between pathological responses and overall survival for the patients having basal tumours. Prospective validation in a larger cohort is required to address this issue. In addition, our group showed that basal tumours are sensitive to epidermal growth factor receptor (EGFR) inhibition in vitro and in preclinical models [41]. Sensitivity to RT of the subtypes remains yet to be investigated, but it has been suggested to be increased in two subtypes: neuroendocrine-like and luminal unstable, which show elevated cell cycle activity and low hypoxia signals [42,43]. At the present time no significant difference has been found in local relapse-free survival between bladder tumour subtypes in MIBC patients treated by TURBT followed by CCRT [44] or RT alone [45].

### 4.2. BCa Cell Line Molecular Subtypes

There is no agreement yet on the molecular classification of BCa cell lines, in particular with regard to the new consensus classification [36]. Our group made a first dichotomy between basal and non-basal cell lines [41]. Then, Earl et al. assigned the subtypes to a series of 40 BCa cell lines [47]. Our group has made more stringent classification [46] For example, five cell lines discussed in this review were identified as “basal” by Earl et al., while classified as “non-luminal, non-basal” by Shi et al. [46]. These non-luminal, non-basal cell lines expressed epithelium to mesenchyme transition markers and do not express E-cadherin. These cells could represent sarcomatoid tumours which is a rare entity in vivo. This phenotype could be present in the initial tumours or acquired in vitro. In vivo, the sarcomatoid tumours are classified mainly in the basal subtype probably due to commonalities in their stroma. However, data from experimental studies on radiosensitisation of relevant models representing different molecular subtypes are sparse. It is worth noting that none of the studies included in this review has discussed the relevance of a molecular subtype of the chosen experimental models. It is in part due to the fact that the consensus was only recently established and that there was an absence of classification of the cell lines until recently.

## 5. BCa Experimental Models in RT Studies

The experimental models and study design should utilise the information available regarding the subtypes of the cell lines and also revisit the information available regarding the patient and the original tumour from which the cell line has been established. BCa experimental models and their molecular features have been recently reviewed by Zuiverloon et al. [48] and rodent models with relevant molecular subtypes have been described by Ruan et al. [49]. Here, we critically discuss the BCa experimental and preclinical models used in association with RT treatment.

### 5.1. BCa Cell Lines

#### 5.1.1. Molecular Subtypes

Selection of BCa cell lines with different mutation status and from different subtypes would better reflect the heterogeneity of MIBC cancer patients. We identified 29 different BCa cell lines used in the experimental studies of RT. We assigned molecular subtypes to the human BCa using a classification recently proposed by our team [46] (Table 1, additional information regarding the mutational status and origin are available in Appendix A). We found that from all the BCa cell lines used in RT studies, 26% are of luminal subtype, which is in contrast to 47% of human tumours considered having luminal features (Figure 1a,b). In total, 21% of cell lines used were of basal subtype (contrary to 33% of human tumours) and the largest part (40%) of BCa cell lines was classified as neither luminal nor basal (Figure 1b). These cells, which express epithelial to mesenchyme transition markers and do not express E-cadherin, are rarely found in vivo. They could represent sarcomatoid tumours or the transient state of tumour cells representing only a fraction of tumours. This transient state could be of importance during the invasion or the metastatic process.

Regrettably, we found that 7% of all in vitro studies and 12% of all in vivo studies have used cell lines that have been identified as cross-contaminated or misidentified (Figure 1b,c, Appendix A).

#### 5.1.2. Gender

Given the difference incidence rates between men and women in BCa, it is important to exclude potential gender bias in the study design and include cell lines from both sexes. BCa is significantly more frequent in males, while the majority of studies (54 of 85 studies identified or 64%) have used RT112 and T24 cell lines which are of female origin (Appendix A). Androgen receptor (AR) signalling could be implicated in the gender disparity of BCa but it remains to be confirmed [51,52]. Furthermore, AR signalling has been recently shown to reduce radiosensitivity [53].

#### 5.1.3. Intrinsic Radiosensitivity

There is a lack of certainty when it comes to the intrinsic radiosensitivity of BCa cell lines as many have been used only in a single study and therefore obtained radiation–response curves have never been reproduced (Appendix A, No. of studies). On the other hand, the wide range of different cell lines for the use of radiosensitisation studies offers opportunity to examine many potential factors influencing radiation response. Only a single dataset comparing intrinsic radiosensitivities of 19 BCa cell lines exists by Yard et al. [54]. Yard et al. used data derived from a single experimental platform and performed analysis using a rigorous statistical methodology. They studied genetic determinants influencing tumour response to DNA damage and influencing tumour survival as assessed by colony forming assays. The radiosensitivity was described by the area under the curve (AUC) and scored from 0 (completely sensitive) to 7 (completely resistant) (Table 2). There was a high heterogeneity among the BCa cell lines (radiosensitivity varying from 1.883 (radiosensitive) to 5.228 (radioresistant)) [54] (Table 2). It is interesting to note that by associating with the molecular subtype, the three most radioresistant cell lines are of basal subtype (AUC ≥ 4.4). However, there were also two basal cell lines reported to be very radiosensitive (AUC < 2.5). Luminal cell lines included in this study had a lower variation of the AUC (2.935–4.126) all being moderately radiosensitive/moderately radioresistant. However, this data set included only five luminal cell lines so more studies having stringent measurements of radiosensitivity are desirable.

### 5.2. BCa Xenografts

The majority of data generated from the RT studies in BCa are currently from BCa xenografts (Table 3); fewer studies have used syngeneic mouse models (Table 4), while none have used patient-derived xenografts (PDX). RT studies using 3D-BCa xenografts have the advantage to gather more clinically-relevant information when compared to in vitro 2D-models. For example, radiosensitivity can be studied in 3D considering hypoxic areas, which have been previously identified as potential cause of therapy failure in BCa. Indeed, a study by Williams et al. found that the hypoxia-targeting prodrug AQ4N efficiently sensitised luminal BCa xenografts to cisplatin-based CCRT (Table 5). However, some interactions such as immune infiltration between tumour and its microenvironment can be limited by the species barrier.

We found that seven (29%) BCa xenografts used were of luminal subtype (Table 3, Figure 1c), while only one study (4%) used the cell line of the basal subtype (Table 3, Figure 1c). Clearly, the BCa xenografts of the basal subtype have been less studied in the RT context. We analysed the in vivo study design whereby different radiation schedules were employed (single larger radiation dose versus fractionated dose delivery schedule) (Table 3). Regrettably, three in vivo studies have used cell lines reported as cross-contaminated (Appendix A).

### 5.3. Syngeneic Models

Evidently, syngeneic mouse models are a suitable choice to test immunotherapy agents in combination with radiation treatment. Furthermore, studies of interactions of tumour cells with endothelial and fibroblastic syngeneic cells are more relevant than in a xenogeneic model. We identified seven studies using syngeneic mouse models (Table 4). The three cell lines used in these studies were all chemically induced (detailed in [48]) and resemble the human basal/squamous subtype [41,49]. We noted that all of these models were heterotopic, excluding the assessment of treatment-induced bladder toxicity. Results of most of the studies are discussed further.

## 6. Use of CT Agents in Combination with RT in Experimental Studies for BCa Treatment

### 6.1. Cisplatin

Cisplatin is currently the most widely used radiosensitising agent in MIBC, supported by a randomised clinical trial [3]. According to the NCCN Clinical Practice Guidelines in Oncology, the most comprehensive guidelines for treatment of oncological patients in the United States, the recommended radiosensitising regimen for locally advanced or metastatic BCa is a combination of cisplatin and gemcitabine [78]. However, it should be emphasised that this combination has not been investigated in our identified experimental and preclinical studies.

In our literature analysis, we found only two in vitro and four in vivo (preclinical) studies investigating cisplatin and RT in BCa (Table 5). Weldon et al., for the first time, used cisplatin in combination with RT in a murine BCa model and compared different administration schedules and included also two other drugs for comparison [18]. They found that the concomitant administration of cisplatin and RT was toxic, but when cisplatin was used as adjuvant therapy after completion of RT, synergistic effect was produced, but another CT drug cyclophosphamide was more effective in terms of growth delay [18]. Further, a study by Kyriazis et al. observed the most synergistic effect when cisplatin was given on days 3 and 6 post-radiation using human BCa xenograft model (SW-800, not classified cell line) [79]. In an in vitro study, Bedford et al. demonstrated that radiation-resistant cell lines are more sensitive to cisplatin and radiation compared to wild-type human BCa cell lines [80]. Kawashima and colleagues investigated whether CCRT response can be predicted using expression of excision repair cross-complementing group 1 (ERCC1) and found that its downregulation improved the effect of CCRT, but not of cisplatin alone in vitro [31]. Two further studies have used mouse BCa xenografts. Yoshida and colleagues investigate the prospect to improve CCRT response by using heat shock protein (HSp90) inhibitors in non-luminal, non-basal mouse BCa xenograft (UMUC3) while evaluating the effect on normal human urothelial (NHU) cells in vitro [69]. They found greater BCa sensitisation to cisplatin-based CCRT after low-dose Hsp90 inhibitor treatment than with the combination of trastuzumab (HER2 blocking antibody) or LY294002 (PI3K inhibitor). A few sensitising effects of NHU to CCRT were found [69]. Furthermore, Williams et al. found that Hypoxia-Activated ProDrug AQ4N increased the efficacy of RT alone and cisplatin-based CCRT in vivo [81].

### 6.2. Gemcitabine

Results from eight phase I-II trials concluded that there is strong evidence that CCRT regimens with concurrent gemcitabine are feasible and well tolerated in BCa [82]. Prospective randomised controlled trials are ongoing to definitively assess the efficacy of gemcitabine-based CCRT for MIBC. From the experimental studies identified in our literature search, two have made use of gemcitabine as a radiosensitiser in vitro, while two have used preclinical mouse models in vivo (Table 6).

There have been some conflicting results among the early studies of the use of gemcitabine in BCa in vitro models. In 2003, Fechner and colleagues showed no effect of radiosensitivity of gemcitabine in four different BCa cell lines (RT112, RT4, T24 and TCC-SUP) with differing p53 status [83]. In contrast, Pauwels et al. demonstrated correlation between gemcitabine-induced S phase block resulting and sensitization to RT of a BCa cell line ECV304 and cell lines from other cancers in vitro [84]. It is worth noting that the cell line used has been recognised as being contaminated by another BCa cell line (T24, non-luminal, non-basal, Appendix A. These studies did not use colony forming assay to investigate radiosensitisation, which is considered the gold standard for assessing RT efficiency. Another study used colony forming assay to compare gemcitabine radiosensitising effect in related bladder cancer cell line MGHU1 and its radiosensitive subclone S40b [85]. They demonstrated that gemcitabine is an effective radiosensitiser in these BCa cell lines, with greater sensitisation in the radioresistant parental line [85]. Interestingly, MGHU1 cells did not show S-phase accumulation, which is the suggested radiosensitisation mechanism, but its subclone S40b did, despite both being radiosensitised by gemcitabine, indicating that S-phase accumulation is unlikely to be a major mechanism of radiosensitisation by gemcitabine [85]. However, also the MGHU1 cell line has been reported as contaminated by T24 cell line (Appendix A) [86].

Further, Choudhury et al. used gemcitabine in combination with another targeted agent (imatinib). Imatinib (inhibitor of c-ABL, c-KIT, and platelet-derived growth factor receptor (PDGFR) tyrosine kinases) was found to be a sensitising agent to RT and gemcitabine-based CT treatment using RT112 luminal cell line in vitro (alongside a prostate cancer cell line in vitro and in vivo), concluding that imatinib can sensitise tumour cells to DNA damaging agents and induce mitotic catastrophe [87]. Two studies from Anne Kiltie team at the University of Oxford have investigated gemcitabine in vivo (Table 6) [55,88]. In the first one, Kerr et al. demonstrated that gemcitabine-resistant Calgem heterotopic xenografts were responsive to the combination of gemcitabine and irradiation [88]. In the second study, Groselj et al. showed that gemcitabine + RT resulted in more acute and late intestinal toxicity than HDAC inhibitor panobinostat + RT [55].

### 6.3. In Vivo Study Reporting/Design

We noted that the in vivo study design of studies using cisplatin and/or gemcitabine have been very few and heterogeneous. Different radiation schedules have been employed (single larger radiation dose vs. fractionated, more clinically relevant schedule). The source of irradiation was different (X-rays and gamma rays) and different dose rates were reported. It has been shown that a variation of X-ray energy and the dose rate can impact the relative biological effectiveness (RBE) both in cells in vitro and in vivo [89,90]. From the studies using gemcitabine and cisplatin, three studies had not reported the dose rate. Different choice of the radiation delivery schedule will prevent a direct comparison of the studies.

## 7. Targeted Agents to Improve RT Response in BCa

Following the studies of classical CT drugs, multiple agents have been tested as potential radiosensitisers since 2000. Nineteen studies using BCa xenografts (Table 7) and six studies of syngeneic mouse tumour models (Table 8) have reported a significant radiosensitisation. These studies are very heterogeneous, testing diverse agents ranging from RTK inhibitors, epigenetic modifiers, hypoxia- or angiogenesis targeting molecules, among others. Below, we comment on the data of few selected studies.

### 7.1. Epidermal Growth Factor Receptor

Receptor tyrosine kinases (RTKs) are frequently differentially expressed between normal and cancerous tissue. They mediate pro-proliferative, pro-survival pathways as well as DNA repair pathways, the activation of which could ultimately protect cancer cells from radiation-induced cell death. Furthermore, radiation-induced activation of several RTKs has been reported and belongs to the earliest events in response to DNA damage [91]; reviewed by [92]).

Epidermal growth factor receptor (EGFR) overexpression has been found in up to 70% of BCa tumours [93]. The team of François Radvanyi has been studying the role of several RTKs in BCa and has identified basal subtype BCa cell line dependency to EGFR when not mutated for a RAS family members [41,70]. EGFR is the most-studied RTK in the field of radiation oncology as it was the first RTK to be shown to be activated with RT [91,92]. A clinical trial combining EGFR blocking monoclonal antibody with RT versus RT alone significantly improved overall survival in head and neck squamous cell carcinoma patients [94].

In RT studies of BCa, Domniguez-Escrig et al. observed radiosensitising effect of the use of the EGFR inhibitor Gefitinib in vitro in two BCa cell lines. However, Gefitinib alone did not cause growth delay in the luminal RT112 xenograft in vivo, but was validated using the basal 253J B-V xenograft [95]. Colquhoun et al. demonstrated that radiation induced activation of EGFR and MAPK and Akt downstream effectors in two BCa cell lines. Further, Gefitinib + RT induced significant growth delay in non-luminal, non-basal J82 cell line xenografts compared to single treatment [70].

### 7.2. Chromatin Modifiers/Epigenetic Regulators

High levels of histone deacetylases (HDACs) have been detected in BCa tumours [96]. HDAC inhibitors have been tested alone in clinical trials in advanced solid tumours, including BCa; however, high reported toxicities to normal tissue have impeded their progress to clinics for example of the HDAC inhibitor mocetinostat [97]. The team of Anne Kiltie at Oxford University has been studying other HDAC inhibitors as potential radiosensitisers in BCa experimental and preclinical models and have found promising results using pan-HDAC inhibitor panobinostat [55] and more selective HDAC inhibitor romidepsin [57]. In addition, no increase in acute or late toxicity following mouse pelvic irradiation has been reported in [55].

### 7.3. Radio-Immunotherapy

In BCa preclinical models, immune checkpoint inhibitors have been used with RT only in one study using an anti-PD-L1 antibody [72] (Table 8). Wu et al. demonstrated that RT upregulated PD-L1 expression in BCa tumour cells, correlating with radiation dose. Using heterotopic MB49 syngeneic mouse models, PD-L1 blockade induced a longer tumour growth delay following irradiation [72]. Bacillus Calmette-Guérin (BCG) bladder instillation is commonly used after local tumour resection for patients with superficial bladder cancer. In addition, BCG bacteria-induced immune response has been studied in BCa to improve response to RT [75]. Invasive murine BCa cell line MB49-I was cultured in monolayers in 2D, in spheroids in vitro in 3D and inoculated in vivo in the syngeneic mice. BCG pre-treatment radio-sensitised spheroids, while no effect was shown in monolayers. In vivo, BCG improved the local response to RT and decreased the presence of lung metastasis. The combined BCG+RT treatment also resulted in abscopal effect, where second tumour development in the opposite flank was completely rejected, compared to the untreated or RT only arms [75].

## 8. Suggestions to Improve the Design of Future Experimental BCa Studies: New Agents and Relevant Models

In order to improve RT, its therapeutic index (i.e., the ratio: antitumour efficiency/toxic effects on surrounding healthy bladder tissues) must be increased. Modern radiotherapy has advanced spatial targeting of clinical tumour volume, including whole bladder and pelvic nodes while reducing side effects to pelvic organs other than the bladder. However, since the whole bladder is included in the clinical target volume (CTV), only strategies aiming at radiosensitising tumour tissue selectively and not or to a lesser extent of the normal bladder wall, will improve therapeutic index of RT.

### 8.1. Molecular Subtype Consideration

Gemcitabine and cisplatin are effective radiosensitisers, but other agents have shown superior effect in the few comparative studies in vitro and in vivo (Table 5 and Table 6). Unfortunately, only one luminal cell line (RT112) and no basal cell line have been used in vivo to evaluate cisplatin- or gemcitabine-based CRT, thus not allowing the study of the differential effect observed in clinics between tumour response to CT of basal and luminal subtypes (where basal tumours are shown to be better responders). It would be important to compare CCRT/RT response in models of different subtypes and study the underlying mechanisms. For that, patient-derived xenografts (PDXs) are relevant models as they can conserve in mice the subtype of the original human tumour [99]. Briefly, PDX establishment consists of engrafting a fragment of a patient tumour directly into an immunocompromised animal, and then maintaining it through passaging from animal to animal, avoiding the in vitro selection and allowing one to conserve the initial histology. In future studies, using a PDX model would eliminate the non-luminal non-basal subtypes that most of the BCa cell lines used in RT studies are representing, but which do not clearly represent human tumours. Despite the possible tumour loss of distinct molecular features over time, PDX models in BCa warrant future efforts to be used in RT studies.

### 8.2. New Targeted Agents

In BCa, several experimental studies have shown the therapeutic efficacy of RTK inhibition other than EGFR. Fibroblast Growth Factor Receptor 3 (FGFR3) has been extensively studied in BCa due to its frequent mutations/translocations in the BCa, driving oncogenic dependency (more than 65% of NMIBCs and 15% of MIBCs) [100,101,102]. Pan-FGFR inhibitor has been recently shown to radiosensitise tumours in head and neck squamous cell carcinoma xenografts and PDX [103]. In the context of BCa subtypes, 50% of luminal BCa tumours harbour an FGFR3 alteration and therefore could potentially benefit the most from anti-FGFR3+RT treatment.

Recently, TYRO3, a member of the TAM family of RTKs (comprising TYRO3, AXL and MERTK) has been identified as a potential target in the BCa [104]. TYRO3 overexpression has been reported in 50% of MIBC and results in TYRO3-dependency for growth of BCa cancer cell lines [104]. However, FGFR3 and TYRO3 have never been explored as targets for radiosensitisation in BCa.

Using syngeneic models allows investigating the therapeutic index of different radiosensitising agents including the effect on tumour cells but also on immune response and other interactions restricted by species barrier. For example, inhibition of the TAM receptors could improve RT efficacy by increasing directly radiation-induced tumour cell killing and also by promoting innate immunity [105,106]. Currently, in the field of radiation oncology, there are many studies in syngeneic mouse models exploring different hypofractionated RT regimens in combination with immune checkpoint inhibitors, such as anti-PD-L1 and anti-T cell immunoreceptor with Ig and ITIM domains (TIGIT) in colorectal cancer [107]. The underlying interest is to combat PD-L1 expression which has been shown to be upregulated upon RT [108]. Further, also abscopal effect is being investigated and it has been shown that RT can promote a response of lung cancer to cytotoxic T-lymphocyte 4 (CTLA-4) blockade [109]. There is clinical evidence that basal subtype benefits from early aggressive management with CT agents and would benefit from T cell modulators (i.e., targeting CTLA-4) and EGFR, NFκB, Hif-1α/VEGF, and/or Stat-3-targeted agents will also be active within this subtype [110]. Such preclinical studies are needed in BCa to improve the RT-immunotherapy modalities.

### 8.3. Pelvic Toxisity Assessment

Experimental devices dedicated to accurate mice irradiation are currently available, which includes high resolution computerised tomography scanner for imaging, warmed beds to keep the model at a physiological temperature during longer irradiating sessions and possibility to target CTV more accurately. This allows studying the side effects of the RT on the bladder, giving the opportunity to compare acute radiation-induced toxicity versus long-term radiation-induced side effects such as fibrosis on the bladder. Although toxicity of normal tissue is one of the main limitations of use of RT in BCa, only two studies have considered pelvic toxicity. First, the study assessed pelvic irradiation-induced intestinal toxicity [56]. Second was an in vitro study using normal human urothelium (NHU) cells [69]. NHU cultures are relevant models to study bladder toxicity in vitro [111]. NHU cultures allow studying the impact on the normal urothelial cell proliferation from the RT or combined treatments as a first step before investigations in vivo. In addition, NHU can be differentiated into the non-proliferative phenotype, which is the physiological state of NHU cells in the human bladder [112,113]. Differentiated NHU cell monolayers would be a more relevant model to study radiation-induced toxicity in vitro.

### 8.4. Use of Orthotopic Mice Models

Currently, there is no preclinical study published using syngeneic or xenogeneic orthotopic graft in the field of BCa RT. However, Jäger et al. have developed a high-precision approach consisting of ultrasound-guided tumour cell inoculation within the bladder wall [114]. Another interesting model is the UPII-SV40T transgenic mouse model which expresses the SV40 large T antigen specifically in the urothelium and reliably develops BCa [115]. In addition, there is a simple chemically induced BCa model, developed by daily exposure to BBN (N-butyl-N-(4-hydroxybutyl)-nitrosamine) in drinking water (0.05%). Around 12 weeks of exposure to BBN result in development macroscopic lesions [116]. At the present time, all of these orthotopic models have never been used in preclinical RT studies.

### 8.5. Humanised Mouse Models

In recent years, humanised mouse models have been more widely used in the field of immunology. For example NOD scid gamma (NSG) highly immunodeficient mouse model have no B-, T-, and natural killer (NK) cells, and therefore allow the engraftment of tumour and immune cells of human origin [117,118]. Indeed, a recent study using this model established BCa xenografts in this humanised system and observed significant tumour growth delay using a pan-PI3K inhibitor in tumours bearing a PIK3CA mutation. Furthermore, pan-PI3K-treated PIK3CA-mutated BCa tumours were sensitive to PD-1 blockade. These results showed potential of combination of PI3K inhibitors with immune checkpoint inhibitors to overcome resistance to immune checkpoint inhibitors [119]. It would be interesting to include additional treatment arms combining such strategies with RT in the future.

## 9. Methods

Following the Preferred Reporting Items for Systematic Reviews and Meta-Analyses (PRISMA) guidelines for systematic reviews [120], we conducted a literature search limited to the database of PubMed. We used the following search string (((radiotherapy) OR (radiation therapy) OR (irradiation) OR (radiation) OR (electromagnetic radiation) OR (electromagnetic irradiation) OR (radiosensitisation) OR (radiosensitivity) OR (radioresistance) OR (radiosensitization) OR (radiation toxicity)) AND ((bladder cancer) OR (urinary bladder neoplasms) OR (Bladder Tumour) OR (urinary bladder) OR (urothelial carcinoma) OR (urothelium)) AND ((cell line) OR (xenograft) OR (syngeneic) OR (preclinical model) OR (pre-clinical model) OR (orthotopic) OR (cells) OR (NHU) OR (normal human urothelium) OR (urothelial cells) OR (mouse) OR (rodent)) NOT ((review) OR (case report) OR (systematic review) OR (meta-analysis))). All of the search was conducted between September and October 2020 without time restrictions. Relevant studies were identified between 1979 and 2020 (until October 31).

Initial study identification was carried out independently by L.S. and F.M. using the search tool in the PubMed with the following inclusion criteria: studies in English, studies with available abstracts, peer-reviewed journal articles only, not reviews.

The titles and abstracts of the obtained studies were further screened independently by L.S. and F.M. and excluded on the basis of following criteria: (1) no experimental models of bladder cancer used, (2) radiotherapy treatment not used or used as a single treatment. The lists were compared and the publications for which the two reviewers had a disagreement were reviewed together and, when needed, discussed with the third reviewer (F.M.C.). Then, the full text was obtained and assessed for eligibility, excluding only clinical studies and studies using photodynamic therapy. After careful reviewing of each full text article, additional studies were excluded where the experimental model used was only non-human BCa cell lines in vitro. The flow chart showing the numbers of the initial studies identified in PubMed and the steps leading to the final inclusion of 85 studies are depicted in Figure 2.

## 10. Conclusions

Increasing the use of bladder preserving radiation-based treatment needs an improved therapeutic index leading to reduced side effects. Experimental studies are needed to address this issue. This review first identified and analysed all experimental investigations on concurrent combination of radiation with different agents in bladder cancer and then provided suggestions aiding the selection of appropriate cell lines, mouse models, radiosensitising agents and radiotherapy regimen to improve the design of future experimental studies.

## Figures and Tables

**Figure 1 cancers-13-00087-f001:**
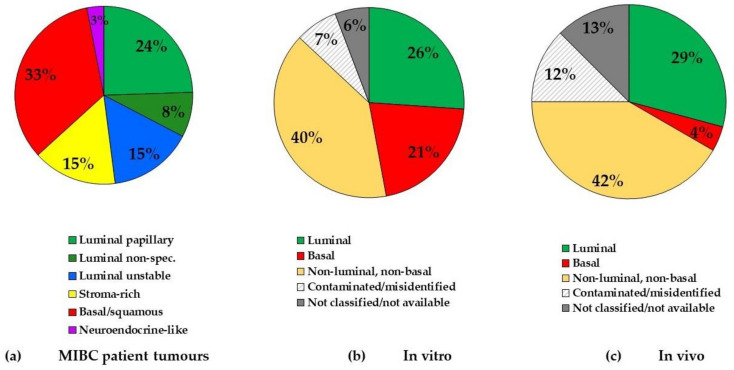
Comparison of muscle-invasive bladder cancer (MIBC) subtype frequency in patient tumours and human bladder cancer (BCa) cell lines used in radiotherapy (RT) experimental studies in vitro and preclinical studies in vivo. (**a**) MIBC tumour subtypes (classified by [36]); (**b**) cell line subtypes used in experimental studies of RT in BCa in vitro (classified by [46]); (**c**) cell line subtypes used in preclinical studies of radiotherapy in BCa in vivo (mice xenografts) (classified by [46]).

**Figure 2 cancers-13-00087-f002:**
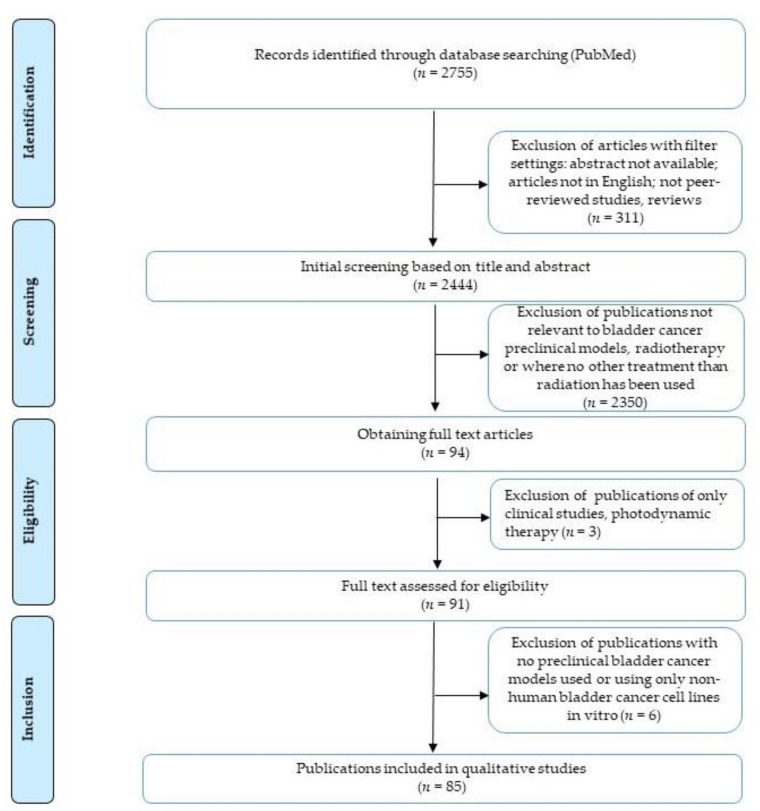
Flow chart showing the identification, screening, evaluation of eligibility and inclusion criteria of publications.

**Table 1 cancers-13-00087-t001:** BCa cell lines used in RT studies.

	Cell Line	Cellosaurus Accession No. [50]	Molecular Subtype [46]
Human	RT112	CVCL_1670	luminal
	SW780	CVCL_1728	
	UMUC5	CVCL_2750	
	UMUC9	CVCL_2753	
	RT4 ^1^	CVCL_0036	
	5637	CVCL_0126	basal
	647V	CVCL_1049	
	HT1197	CVCL_1291	
	HT1376	CVCL_1292	
	KU19-19	CVCL_1344	
	UMUC6	CVCL_2751	
	VMCUB1	CVCL_1786	
	253J B-V	CVCL_7937	non-luminal, non-basal
	639-V	CVCL_1048	
	J82	CVCL_0359	
	KK47	CVCL_8253	
	T24	CVCL_0554	
	TCC-SUP	CVCL_1738	
	UMUC3	CVCL_1783	
	CAL29	CVCL_1808	n/c
	NTUB1	CVCL_RW29	n/a
	OBR	n/a	
	SW-800	CVCL_A684	
	UCRU-BL13	CVCL_M873	
	UCRU-BL17	CVCL_M007	
	UCRU-BL28	CVCL_4904	
Mouse	MB49	CVCL_7076	basal (mouse) ^2^
	MB49-I	CVCL_VL62	
	MBT2	CVCL_4660	

^1^ RT4 cell line originates from re-occurring human transitional cell papilloma; ^2^ mouse cell lines have been classified by [49]. Abbreviations: n/a, information not available (classification have not been applied); n/c, not categorised (not coherent classification depending on the dataset used).

**Table 2 cancers-13-00087-t002:** Intrinsic radiosensitivity of a panel of BCa cell lines reported by Yard et al., 2016 [54].

Cell Line	AUC ^1^ [54]	Molecular Subtype [46]
HT1376	5.228	basal
HT1197	4.449	basal
VMCUB1	4.412	basal
KMBC2	4.126	luminal
TCCSUP	3.539	non-luminal/non-basal
KU1919	3.503	basal
647V	3.374	basal
BC3C	3.362	basal
UMUC1	3.346	luminal
SW1710	3.309	n/c
UMUC3	3.231	luminal
J82	3.198	non-luminal/non-basal
RT112	3.038	luminal
RT4 ^2^	2.935	luminal
UBLC1	2.914	n/c
JMSU1	2.792	non-luminal/non-basal
5637	2.473	basal
T24	2.366	non-luminal/non-basal
SCABER	1.883	basal

^1^ AUC: Area under the curve; ^2^ RT4 cell line originates from re-occurring human transitional cell papilloma. Abbreviations: n/c, not categorised (not coherent classification depending on the dataset used).

**Table 3 cancers-13-00087-t003:** Overview of human BCa cell line xenograft models used in radiosensitisation studies.

Subtype [46]	Cell Line	IR Regimen	Radiosensitising Agent	Class	Nude Mice Genetic Background (Gender)	Initial Tumour Size (mm^3^) ^1^	Study Follow-Up (Days) ^2^	Ref.
luminal	RT112	4 × 5 Gy	Panobinostat (vs. gemcitabine)	HDAC inhibitor	(Unknown strain) (F)	100	10–60	[55]
		2 × 5 Gy	AQ4N (banoxantrone) (vs. cisplatin)	DNA intercalator and Topoisomerase II inhibitor	CBA (F)	240–280	10–60	[56]
		1 × 6 Gy	Romidepsin	HDAC inhibitor	CD1 (F)	50	25	[57]
		1 × 6 Gy	Low-/soluble high-/insoluble high- and mixed high-fibre diets	Diet	CD1 (F)	50	42	[58]
	RT4	1 × 5 or1 × 15 Gy	Photofrin II	Photosensitiser	(Unknown strain) (F)	2.6–3.0	15	[59]
		1 × 2 Gy	Caffeine	DNA Damage Response inhibitor	BALB/c (M)	30–75	0 ^3^	[60]
	SW780	2 × 5 Gy	siTUG1	siRNA	(Unknown strain) (M)	100	21	[61]
basal	5637	2 × 2 Gy	Sulfoquinovosylacylpropanediol	Synthetic sulfoglycolipid	BALB/c Slc (M)	100–300	33	[62]
Non luminal/non basal		1 × n/a Gy	shRNF8	shRNA	BALB/c (M)	100–150	30	[63]
		1 × 6 Gy	Chloroquine	Other	BALB/c (F)	∼200	25	[64]
		1 × 6 Gy	Nanoparticles (chloroquine conjugated)	Nanoparticles	(Unknown strain) (n/a)	150	16	[65]
		1 × 6 Gy	LY294002	TKI	Ncr-nu/n (F)	300–400	40	[66]
		1 × 6 Gy	FTI-276 or L744832	Farnesyltransferase inhibitors	Ncr-nu/n (n/a)	58	80	[67]
	UMUC3	2 × 3 Gy	shHMGB1	shRNA	(Unknown strain) (F)	n/a	21	[68]
		1 × 12 Gy	17-AAG or 17-DMAG/Trastuzumab/LY294002	Hsp90 inhibitors/ monoclonal antibody/TKI	BALB/c (M)	1000	12	[69]
		2 × 2 Gy	Flutamide/shAR	Antiandrogen/shRNA	NOD-SCID (M)	30	12	[53]
	J82	1 × 5 Gy	Gefitinib (“Iressa”, ZD1839)	TKI	BALB/c (n/a)	100	n/a	[70]
n/a	KK47	1 × 4 Gy	Ad-RSV-CD+5-FC	A recombinant adenovirus vector	BALB/c (n/a)	n/a	n/a	[71]

^1^ The initial size of the tumour is defined as the size of the tumour at the start of the RT or combination treatment (Day 1). ^2^ The minimum follow-up for the non-treated control was used to compare the growth of the xenografts. ^3^ In this study, the mice were sacrificed immediately after the treatment delivery. Abbreviations: AR, androgen receptor; HDAC, histone deacetylase; HMGB1, high mobility group box 1; Hsp90, heat shock protein 90; n/a, information not available; RNF8, ring finger protein 8; TKI, tyrosine kinase inhibitor; TUG1, taurine upregulated gene.

**Table 4 cancers-13-00087-t004:** Overview of mouse BCa syngeneic models used in radiosensitisation studies.

Cell Line	IR Regimen	Radiosensitising Agent	Class	Mouse Background (Gender)	Initial Tumour Size (mm^3^) ^1^	Study Follow-Up (days) ^2^	Ref.
MB49	1 × 12 Gy	PD-L1 blocking antibody	Immunotherapy	C57BL/6 (F)	500	27	[72]
MB49	2 × 5 Gy	Glycyrrhizin	HMGB1 inhibitor	C57BL/6 (M)	Once palpable	7	[73]
MB49,MB49-I	6 × 3 Gy	Silybin (Sb)	Flavonoid	C57BL/6J (n/a)	50	30	[74]
MB49-I	6 × 3 Gy	Bacillus Calmette-Guérin (BCG)	Immunotherapy	C57BL/6J (n/a)	50	21	[75]
MBT-2	1 × 15 Gy	Lapatinib	TKI	C3H/HeN (F)	162	21	[76]
MBT-2	1 × 15 Gy	Afatinib	TKI	C3H/HeN (F)	162	21	[77]
MBT-2	5 × 4 Gy	Cisplatin, doxorubicin hydrochloride (adriamycin), cyclophosphamide	CT	CsH/Hej (n/a)	6	60	[18]

^1^ The initial size of the tumour is defined as the size of the tumour at the start of the RT or combination treatment (Day 1). ^2^ The minimum follow-up for the non-treated control was used to compare the growth of the xenografts. Abbreviations: CT, chemotherapy; HMGB1, high mobility group box 1; PD-L1, programmed death ligand 1; TKI: tyrosine kinase inhibitor.

**Table 5 cancers-13-00087-t005:** Overview of preclinical studies using Cisplatin in BCa in vivo in combination with RT.

	Yoshida et al., 2011 [69]	Williams et al., 2009 [56]	Kyriazis et al., 1986 [79]	Weldon et al., 1979 [18]
Cell lines	UMUC3 (non-luminal, non-basal)	RT112 (luminal)	SW-800 (not classified)	MBT-2
Source/ Dose rate (Gy/min)	X-rays (225 V)/ 0.83 Gy/min	X-rays (230 kV)/ 2 Gy/min	X-rays (250 kV)/ 1.23 Gy/min	X-rays (250 kV)/ n/a
IR dose and fractionation	5 × 2 Gy	5 × 2 Gy	1 × 10 Gy	5 × 4 Gy
Cisplatin dose	3 mg/kg (administered once)	2 mg/kg (administered once)	5 mg/kg once on each specified day before or after radiation	3 mg/kg once a week (3 weeks)
Treatment arms	Hsp90 inhibitors (17-AAG or 17-DMAG) Trastuzumab, LY294002	AQ4N (banoxantrone)	-	Doxorubicin hydrochloride (Adriamycin), cyclophosphamide
Normal tissue toxicity	Yes (NHU in vitro)	-	-	-

**Table 6 cancers-13-00087-t006:** Overview of studies using Gemcitabine in BCa preclinical studies in combination with RT.

	Groselj et al., 2018 [55]	Kerr et al., 2014 [88]
Cell lines	-	CALgem
Source/ Dose rate (Gy/min)	X-rays (220 kV)/ n/a	gamma (^137^Cs)/ 1.7 Gy/min
IR regimen in vivo	1× 10, 12, or 14 Gy (acute toxicity) 5 × 5 Gy (late toxicity)	5 × 2 Gy
Gemcitabine dose	single 100 mg/kg injection	single 100 mg/kg injection
Normal tissue toxicity	yes (intestinal)	-

**Table 7 cancers-13-00087-t007:** Targeted agents used as radiosensitisers in preclinical studies using BCa xenografts.

Class	Name	Target	Cell Line Subtype (According to [46])	Year	Ref.
TKI	Gefitinib (ZD1839)	EGFR	J82, non-luminal, non-basal	2007	[70]
	Afatinib, Erlotinib	EGFR/HER2, EGFR	NTUB1, class n/a	2015	[98]
PI3K	LY294002	PI3 kinase	T24, non-luminal, non-basal	2003	[66]
Epigenetic modifiers	Panobinostat	HDAC (histone deacetylase)	RT112, luminal	2018	[55]
	Romidepsin	HDAC (histone deacetylase)	RT112, luminal	2020	[57]
Heat shock protein inhibitors	17-AAG or 17-DMAG	Hsp90	UMUC3 non-luminal, non-basal	2011	[69]
Farnesyltransferase inhibitors	FTI-276 and L744832	Farnesyltransferase	T24, non-luminal, non-basal	2000	[67]
Hypoxia	AQ4N	Hypoxia	RT112, luminal	2009	[56]
Angiogenesis	SQAP	Angiogenesis	5637, basal	2016	[62]
Other	Chloroquine	Autophagy	T24, non-luminal, non-basal	2018	[64]
	HSA-MnO2-CQ nanoparticles	Autophagy	T24, non-luminal, non-basal	2020	[65]
	Ad-RSV-CD+5-FC	-	KK47, non-luminal, non-basal	2003	[71]
	shRNF8	DNA Damage Response	T24, non-luminal, non-basal	2016	[63]
	Caffeine	DNA Damage Response	RT4, luminal	2015	[60]
	siTUG1	HMGB1	SW780, luminal	2017	[61]
	shHMGB1	HMGB1	UMUC3 non-luminal, non-basal	2016	[68]
	Flutamide/shAR	AR	UMUC3 non-luminal, non-basal	2018	[53]
	Photofrin II	Angiogenesis	RT4, luminal	2001	[59]
	Low-fibre, soluble high-fibre, insoluble high-fibre, and mixed soluble/insoluble high-fibre diets	Metabolism	RT112, luminal	2020	[58]

Abbreviations: 17-AAG: 7-N-allylamino-17-demethoxygeldanamycin; 17-DMAG: 17-Dimethylaminoethylamino-17-demethoxygeldanamycin; AQ4N: banoxantrone dihydrochloride, topoisomerase II inhibitor; AR: androgen receptor; HER2: human erbB-2 receptor; HMGB1: high mobility group box 1; HSA-MnO2-CQ: MnO2 and chloroquine in human serum albumin (HSA)-based nanoplatform; RNF8: ring finger protein 8; SQAP: sulfoquinovosylacylpropanediol; TKI: tyrosine kinase inhibitor; TUG1: taurine-upregulated gene 1.

**Table 8 cancers-13-00087-t008:** Targeted agents used as radiosensitisers in preclinical studies using syngeneic mice.

Class	Name	Target	Cell line (Subtype According to [46])	Ref.
TKI	Afatinib	EGFR/HER2	MBT-2, mice cell line (basal)	[77]
	Lapatinib	PDGF-R	MBT-2, mice cell line (basal)	[76]
HMGB1 inhibitor	Glycyrrhizin	HMGB1	MB49, mice cell line (basal)	[73]
Flavonoid	Silybin		MB49, MB49-I, mice cell lines (basal)	[74]
Immune checkpoint inhibitor	Anti-PD-L1 antibody	PD-L1	MB49, mice cell line (basal)	[72]
Non-specific immune stimulator	Bacillus Calmette-Guérin (BCG)	Immune system	MB49-I, mice cell line (basal)	[75]

Abbreviations: EGFR, epidermal growth factor receptor, HER2, human erbB-2 receptor; HMGB1, high mobility group box 1; PDGF-R, platelet-derived growth factor receptor; PD-L1, programmed death ligand 1; TKI: tyrosine kinase inhibitor.

## Data Availability

No new data were created or analysed in this study. Data sharing is not applicable to this article.

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
