# Peer review of "Review of Experimental Studies to Improve Radiotherapy Response in Bladder Cancer: Comments and Perspectives"

_cancers, 2020, doi:10.3390/cancers13010087_

Round 1

Reviewer 1 Report

The authors present a well-made systematic review focused on preclinical studies of radiosensiting
strategies in Bladder Cancer. The review was carried out through a literature PubMed search
following PRISMA guidlines for systematic reviews. The authors make comments on preclinical
model selection, esperimental design and results, providing suggestions to improve forthcoming
studies.

Here I report my few comments:
- Paragraph 2,3 and 4 are too long-winded and they did not focus on the main topic. It would be
easier to make them shorter in order to focus immediately on the topic discussed.
- The literature search is made only on PubMed. Please include in limitations PubMed database.
- Please list in a supplementary table the selected paper and specify level of evidence for each of
them.
- Please specify wich autor performed paper screening and selection

Author Response

Dear Reviewer,

Thank you very much for your insightful comments.

Please see below our responses :

The authors present a well-made systematic review focused on preclinical studies of radiosensiting strategies in Bladder Cancer. The review was carried out through a literature PubMed search following PRISMA guidlines for systematic reviews. The authors make comments on preclinical model selection, esperimental design and results, providing suggestions to improve forthcoming
studies.

Here I report my few comments:
- Paragraph 2,3 and 4 are too long-winded and they did not focus on the main topic. It would be easier to make them shorter in order to focus immediately on the topic discussed.

The paragraphs 2,3 and 4 have been shortened to focus to the preclinical studies. The following details on the historical context has been deleted:

Paragraph 2: Meta-analysis of studies has shown comparable survival outcomes and bladder preservation rates leading to the inclusion of trimodality treatment in the current MBIC treatment guidelines as an equally effective treatment for well selected patients [20]. For RT, two schedules are in common use worldwide: a split-dose regimen with interim cystoscopy is used in the United States [21], whilst a single phase treatment is most widely used elsewhere [4].

Paragraph 3: Bladder response to RT includes early toxicity that occurs within six months post-RT and late bladder toxicity that occurs after six months. Bladder wall and/or tumour movement of up to 4 cm and volume variations of up to 44% have been detected using repeated computed tomography scans during therapy volume, presenting significant challenge to spare surrounding tissue [29,30].

Paragraph 4: This classifier can be applied to data from frozen as well as to paraffin-embedded samples analyzed by different technologies (arrays, RNA-seq, including 3’RNA-seq). The first retrospective study by Choi et al. indicated that patients with basal tumours have the highest benefit from neoadjuvant CT [40]. This was further confirmed in a larger retrospective study by Seiler et al., where no difference was observed between cisplatin-based or alternative CT regimens [42].

- The literature search is made only on PubMed. Please include in limitations PubMed database.

The use of only PubMed as a search database has been mentioned in the Abstract and the Introduction. We further complement the Methods section by adding a precision ‘we conducted a literature search limited to the database of PubMed’.

- Please list in a  supplementary table the selected paper and specify level of evidence for each of
them.

According to our knowledge, the level of evidence is relevant only to clinical trials. Currently, there is no level 1a clinical trial comparing the radical cystectomy and radiation-based bladder sparing treatment or comparing two modalities of chemoradiation. Our aim was not to analyse the clinical trials of chemoradiation in bladder cancer. However, we added a list of clinical studies of concurrent chemoradiation in bladder cancer cited in our review without mentioning the exact level of evidence (Supplementary Table 1).

- Please specify wich autor performed paper screening and selection

We complemented the Methods section by adding the following description : ‘The titles and abstracts of the obtained studies were further screened independently by L.S. and F.M. and excluded on the basis of following criteria: 1) no experimental models of bladder cancer used, 2) radiotherapy treatment not used or used as a single treatment. The lists were compared and the publications for which the two reviewers had a disagreement were reviewed together and when needed, discussed with the third reviewer (F.M.C.).

Yours Sincerely,

Prof. Pierre Verrelle

Dr. Linda Silina

Reviewer 2 Report

I read the works with great interest. It contains a comprehensive discussion of issues related to the response to radiation therapy in bladder cancer. It takes into account the latest aspects such as the molecular classification of bladder cancers and the critical analysis of data obtained by other researchers, in particular the results obtained on cell lines. In my opinion, it is a valuable study with logically separated sections and paragraphs.

Author Response

Dear Reviewer,

Thank you very much for your positive feedback. We are delighted that you found our work valuable and interesting. We made some amendments to the original manuscript following other reviewer advice.

I read the works with great interest. It contains a comprehensive discussion of issues related to the response to radiation therapy in bladder cancer. It takes into account the latest aspects such as the molecular classification of bladder cancers and the critical analysis of data obtained by other researchers, in particular the results obtained on cell lines. In my opinion, it is a valuable study with logically separated sections and paragraphs.

Yours Sincerely,

Prof. Pierre Verrelle

Dr. Linda Silina

Reviewer 3 Report

The presented review by Silina and colleagues comprehensively summarizes the status quo of preclinical achievements and concepts to improve radiotherapy response in bladder cancer during the last decades. The authors describe this ongoing research field in a structured and thorough manner beginning with the background of daily clinical use and limitations of radiotherapy in bladder cancer.  Currently relevant topics like molecular subtyping were considered as well as in depth specification of classical and novel targeted treatment including TKIs and epigenetic modifiers combined with radio-therapeutic approaches. Overall, the presented review is an asset for the scientific community providing an excellent overview pointing finally out both the significance and the lack of preclinical studies so far. However, the description of the methods should be complemented by definitions to ensure correct interpretation of underlying studies. Although the graphical illustration of the review design is clearly presented, I would suggest to specify the following criteria: a) please comment on your procedure to exclude studies without available full-text. Does this mean the authors exclude studies when missing access to the journal was given? This could lead to overlook substantial studies in this area. b) “Preclinical” is a widely used term. From the pharmaceutical point of view preclinical models generally comprises animal studies. However, the authors also consider cell line in vitro models upon their meta-analyses. Thus, I would suggest a clear definition how the authors define “preclinical” in this review.  

Author Response

Dear Reviewer,

Thank you very much for your insightful comments and your positive feedback. We are delighted that you found our work relevant and as an asset for the scientific community.

Please see below our responses:

The presented review by Silina and colleagues comprehensively summarizes the status quo of preclinical achievements and concepts to improve radiotherapy response in bladder cancer during the last decades. The authors describe this ongoing research field in a structured and thorough manner beginning with the background of daily clinical use and limitations of radiotherapy in bladder cancer.  Currently relevant topics like molecular subtyping were considered as well as in depth specification of classical and novel targeted treatment including TKIs and epigenetic modifiers combined with radio-therapeutic approaches. Overall, the presented review is an asset for the scientific community providing an excellent overview pointing finally out both the significance and the lack of preclinical studies so far.

However, the description of the methods should be complemented by definitions to ensure correct interpretation of underlying studies. Although the graphical illustration of the review design is clearly presented, I would suggest to specify the following criteria:

a) please comment on your procedure to exclude studies without available full-text. Does this mean the authors exclude studies when missing access to the journal was given? This could lead to overlook substantial studies in this area.

Thank you for pointing out this limitation in the initial search step of the publications. Following your remark, we repeated the search in PubMed using the provided search string and using the date of 31st October 2020 as the only filter and we identified only 15 publications more than at the initial search reported (2755 vs 2740). In order to ensure that we further investigate these 15 publications, we went through the other steps and updated the Figure 2. These 15 identified publications were excluded at the step 3 (not relevant to bladder cancer preclinical models, radiotherapy or no other treatment than radiotherapy). Please find below a detailed description.

STEP 1: We identified 2755 publications instead of 2740 publications previously identified in our initial search.

STEP 2: Following the application of the following filters: Publications where abstract available and publications available in English language: We identified 2444 publications (exclusion of 311 publications).

STEP 3: Screening Titles and abstracts independently by two reviewers (L.S. and F.M.). Exclusion of 2350 publications not relevant to bladder cancer preclinical models, radiotherapy or where no other treatment than radiation has been used. Where there was disaccord between the two reviewers, the third reviewer was consulted (F.M.C.)

Further the following steps remained unchanged.

We also complemented the Methods section by adding the following description : ‘The titles and abstracts of the obtained studies were further screened independently by L.S. and F.M. and excluded on the basis of following criteria: 1) no experimental models of bladder cancer used, 2) radiotherapy treatment not used or used as a single treatment. The lists were compared and the publications for which the two reviewers had a disagreement were reviewed together and when needed, discussed with the third reviewer (F.M.C.).

b) “Preclinical” is a widely used term. From the pharmaceutical point of view preclinical models generally comprises animal studies. However, the authors also consider cell line in vitro models upon their meta-analyses. Thus, I would suggest a clear definition how the authors define “preclinical” in this review.  

We agree with your comment that ‘preclinical’ studies refer to in vivo animal studies. Therefore we changed the title of our work to ‘Review of Preclinical Experimental Studies to Improve Radiotherapy Response in Bladder Cancer: Comments and Perspectives’ to highlight that not only animal studies, but also other experimental models have been discussed in our work. We specified in the abstract that we define ‘experimental studies’ as including both in vitro and in vivo studies.

We also screened the text and changed the word ‘preclinical’ to experimental when referring to cell line only studies, these changes are tracked in the attached manuscript.

Yours Sincerely,

Prof. Pierre Verrelle

Dr. Linda Silina